

# A new coastal ice-core site identified in Dronning Maud Land, Antarctica, for high-resolution climate reconstructions to the Last Glacial Maximum

Vikram Goel[1], Carlos Martin[2], Kenichi Matsuoka[3], Bhanu Pratap[1], Geir Moholdt[3], Rahul Dey[1], Chavarukonam M. Laluraj[1], Meloth Thamban[1]

[1]National Centre for Polar and Ocean Research, Ministry of Earth Sciences, Goa, India
[2]British Antarctic Survey, Natural Environmental Research Council, Cambridge, UK
[3]Norwegian Polar Research Institute, Norway

*Correspondence to*: Vikram Goel (vikram.goel@outlook.com)

**Abstract.** High-resolution ice cores from the Antarctic Ice Sheet margin are crucial for reconstructing the climate history of Antarctica and the Southern Ocean. Ice-rise summits with stable positions and substantial snow accumulation can be ideal sites for such ice cores. We surveyed two ice rises at 16° E, at the eastern edge of the Lazarev Ice Shelf. Kupol Verbljud (VER) is an isle at the calving front, and Kamelryggen (KAM) is a promontory landward of VER. Radar survey reveals ice thicknesses of 560 m under VER's summit and 525 m under KAM's summit. The long-term stable englacial features, Raymond Arches, are observed in both ice rises, but while VER's arches are tilted, KAM exhibits vertically-aligned arches within its summit, indicating a more stable summit position. We find KAM's summit area better suited for a long ice core, given its gentler bed slope and simpler ice stratigraphy. Surface mass balance derived from dated reflectors show consistent spatial patterns over recent decades. Using a one-dimensional age-depth model we consider the local ice flow as a combination of two extreme cases: diverging divide flow and shear-dominated flank flow. We determine which combination of these flow regimes best reproduces the mapped englacial radar stratigraphy and use it to estimate the age of ice. We conclude that KAM's summit is well-suited for obtaining a high-resolution ice core record beyond the Last Glacial Maximum with expected ~20 kiloyear-old ice at a depth 80 m above the bed where the resolution is expected to be 2.5 a cm$^{-1}$.



## 1 Introduction

Antarctica has experienced several significant climatic events since the Last Glacial Maximum (LGM; 20 ka). These events, such as the Last Deglaciation (18-11 ka; Mayewski et al., 1996), the Antarctic Cold Reversal (14.7-13 ka; Pedro et al., 2016), and the abrupt cooling event 8.2 ka ago (Stager and Mayewski 1997), often exhibit distinct characteristics from comparable events in the Northern Hemisphere (Pedro et al., 2011, Burroughs et al., 2003). For example, the Antarctic Cold Reversal coincided with the warm Bølling-Allerød period in the Northern Hemisphere, while the impact of the 8.2 ka cooling event was

less pronounced in the Southern Hemisphere than in the Northern Hemisphere (Wiersma and Renssen, 2006). The causes underlying these climate contrasts between the two hemispheres remain unclear. These climatic events manifest primarily as changes in temperature and sea level. As Antarctica's low-elevation coastal regions are at the forefront of these changes, they are ideal locations to study the southern polar climate and gain valuable insights into their connection to the global climate.

The coast of Dronning Maud Land (DML) faces the Atlantic sector of the Southern Ocean, which is characterized by

the Weddle Gyre and its strong connections between the ice sheet, the atmosphere, and the global ocean circulation through its deep waters. The Atlantic meridional circulation couples climate changes in the northern hemisphere to this sector of the Southern Ocean (Rahmstorf, 2006). As a pivotal component of the larger carbon cycle, this region significantly influences global climate patterns over timescales ranging from centuries to millennia (Vernet et al., 2019). Despite its importance, there is a lack of continuous millennia-scale, high-resolution climate records from the coast of DML.

Ice core records from the DML coast (elevation below 2000 m a.s.l.) have been used to reconstruct surface mass balance (SMB) (Thomas et al., 2017; Philippe et al., 2016; Vega et al., 2016; Schlosser et al., 2014; Kaczmarska et al., 2004), temperature (Ejaz et al., 2022; Naik et al., 2010; Divine et al., 2009), melt chronology (Dey, 2023; Kaczmarska et al., 2006), sea ice (Ejaz et al., 2021), dust influx (Laluraj et al., 2020) and other climate variables at an annual to decadal timescale. The high resolution of the retrieved records is ensured due to the high surface mass balance across the coast of DML (Vega et al.,

2016; Philippe et al., 2016; Schlosser, 1999). However, since these cores are limited to less than three centuries (Thomas et al., 2017), there is a need to retrieve longer ice cores from this region.

To identify an ideal site for an ice core record in coastal DML, we used five criteria: (I) a simple ice-flow regime to avoid ice-dynamical complexities in the ice core data; (II) minimal or no surface melting, as melting can potentially affect the distribution of water isotopes and trace element chemistry within the ice core; (III) sufficiently old ice, at least 20,000 years or

older; at (IV) sufficiently high resolution to allow for annual layer counting, and (V) proximity to a logistics support hub, in our case the Indian Maitri Station in central DML (12° E, Fig. 1a). These criteria increase the likelihood of obtaining a high-resolution ice core that captures valuable climate information.

Ice rises are grounded ice bodies surrounded by ice shelves or the ocean, located at Antarctic Coast (Matsuoka et al., 2015). Summits of ice rises can be suitable for ice coring as they have negligible horizontal flow, supporting our criteria (I).

Ice rises with stable summit positions for extended periods exhibit uplifted internal layers, allowing access to older ice at shallower depths compared to locations away from the ice divide (Raymond, 1983), aligning with our criteria (III). This





characteristic also results in higher resolution for older records towards the base of the ice rise, supporting our criteria (IV). Furthermore, their proximity to the sea increases the likelihood of extracting valuable records to study sea-ice variability (Ejaz et al., 2021).

Numerous ice rises along the DML coast could potentially serve as deep ice core sites (Goel et al., 2020). Leningradkollen Ice Rise, the closest to Maitri (Fig. 1a), experiences frequent surface melting due to its low elevation (174 m a.s.l.; Dey, 2022), making it unsuitable. The second closest ice rise, Djupranen (DJU), is higher (321 m a.s.l.) and has minimal melt features observed in the past 90 years (Dey et al., 2022). It is of a moderate ice thickness (~420 m; Lindbäck et al., 2020) and has stable Raymond Arches (Goel et al., 2020). However, high SMB over this ice rise (Pratap et al., 2022) limits the

expected ice-core age to the last 10 ka (see Section 4.1). Consequently, we focused our investigation on two other ice rises within Maitri's logistical range: Kamelryggen (KAM) and Kupol Verbljud (VER), situated between the Lazarev and Hugin ice shelves (Fig. 1a).

KAM is a promontory with a ridge extending from the ice sheet to a saddle before rising into a seaward ice dome (Fig. 1b). VER is an isle-type ice rise at the calving front situated seaward on KAM. The two ice rises are of comparable size,

with an area of ~900 km$^2$ (Moholdt and Matsuoka, 2015). KAM has a ridge oriented in the northeast-southwest direction across the summit. VER has a more curved shape, with its ridge changing orientation from a more northeast-southwest direction at its summit to an easterly direction at the northern end. Both ice rises have well-developed domes, and an airborne radar profile over KAM shows ~500 m thick ice (BEDMAP2; Fretwell et al., 2013), implying a long climate history. They represent one of several promontory-isle pairs along the DML coast, potentially originating from a single larger promontory that separated

during glacial retreat (Goel et al., 2020; Favier et al., 2016).

Here, we present the results of extensive glaciological site surveys conducted on KAM and VER in the austral summer of 2021-22. We investigate their glaciological settings, including local bed topography, ice stratigraphy, and SMB. We use this information and an age-depth model to evaluate their suitability and determine an optimal site for obtaining a high-resolution ice core. Our evaluation suggests KAM is the most promising site for reaching ice and climate reconstructions back

to the Last Glacial Maximum (LGM).

## 2 Data and methods

### 2.1. Field surveys and data analysis

During the austral summer of 2021-22, a ground-based field survey was conducted over KAM and VER ice rises. The survey included two distinct radar systems: a deep-sounding system with a 2 MHz antenna frequency (or HF ground-penetrating radar;

terminology as per Schlegel et al., 2023) and a shallow-sounding system (Pusle Ekko) with a 200 MHz antenna frequency for shallow sounding (VHF ground-penetrating radar). The primary objectives of these radar surveys were to map the bed topography and englacial stratigraphy using deep-sounding radar profiling and to map the shallow englacial stratigraphy to



assess the SMB patterns using shallow-sounding radar profiling. The radar systems were towed on sledges behind snowmobiles at approximately 8–10 km hr⁻¹. Both measurements were made concurrently.

95        The radar surveys were designed as a square grid incorporating 10-km long parallel profiles (Fig. 1b, c). The surveys were centred around the ice rise summits and oriented to include several profiles crossing each ice rise's ridge. An along-ridge profile connected these cross-ridge profiles, extending over the saddle at KAM while tracing the ridge's curvature at VER. Over KAM, the profile across the summit was further extended by 5 km on each side to capture the larger topographical and SMB variations across the ridge. Total radar profiling was about ~200 km for each radar system made on each ice rise.

100        The radar survey measurements were geotagged using GNSS precise-point positioning with Trimble NetR9 receivers, mounted on the snowmobile towing the radar system. The resulting measurements yielded an average spacing of approximately 3 m for the deep-sounding radar and 0.4 m for the shallow-sounding radar. Data processing involved applying a dewow filter, an Ormsby band-pass filter, and depth-variable gain functions (Goel et al., 2017). Ice thickness was calculated using a radio-wave propagation speed of 169 m $\mu$s$^{-1}$, incorporating a firn correction of 7–10 m applied to account for faster propagation in
the firn layer. This correction was estimated along the survey profile using a steady-state firn density model (Herron and Langway, 1980), constrained by density data obtained from a firn core drilled at each summit and along-track SMB derived from radar reflectors dated with these firn cores (see last paragraph in Sec. 2.1). The firn-correction was determined from the calculated density profiles using the speed-density relation by Kovacs et al. (1995).

        The two firn cores from the summits of KAM and VER were 16.3 m and 14.6 m deep, respectively. The cores were
labelled and stored under refrigeration (−20 °C) and shipped to the National Centre for Polar and Ocean Research (NCPOR), India. The firn core samples were kept in the -20 °C cold room at NCPOR until further processing. The firn cores were manually decontaminated by chipping a thin outer layer using microtome blades and sub-sampled at 5 cm intervals for stable water isotope and major-ion analysis. The water isotopes were analyzed using a Triple Isotopic Water Analyzer (Emanuelsson et al., 2015). The sodium (Na$^{+}$) and sulphate (SO$_4^{2-}$) ions were analyzed using an ion-exchange chromatograph (Dionex 5000)
to quantify non-sea-salt sulphate ($nss$SO$_4^{2-}$ = SO$_4^{2-}$ − 0.252Na$^{+}$). Peaks in these values were used as markers of historical volcanic eruption events to improve chronological constraint (Laluraj et al., 2011; Thamban et al., 2010). The detection limits of the ion chromatography system for the major ions measurements were up to 2 ppb, and precision on several replicate analyses of samples and standards was better than 5%. The age control for VER and KAM firn core is based on complementary methods of annual layer determination using the summer maxima in δ¹⁸O values and non-sea salt sulphate markers of volcanic
eruptions. The resulting maximum ages are 24 years for the VER firn core and 33 years for the KAM firn core. Firn core densities were measured at a 5 cm resolution, determined by dividing the mass by the volume of each sample.

        To study the spatial distribution of SMB over the ice rises, we tracked the deepest prominent radar reflector visible in the shallow radar stratigraphy and dated it using the firn core. Because the tracked reflector is shallow (the ratio of the reflector depth and ice thickness is approximately 0.03), vertical strain effects on its depth are negligible (the shallow layer
approximation, Waddington et al., 2007). To estimate the mass above the reflector, we assume no lateral variations in the





vertical firn density profile and use the density profiles from the firn cores at each ice rise. Dividing this mass by the reflector's age provides the SMB.

## 2.2. Ice-flow model

To estimate the age of ice around the ice rise summits, we used a simplified 1-D map view age-depth model along the radar surveys. Similar simplified models have been used to estimate ice's age at other drilling sites like Vostok (Parrenin et al., 2001; Parrenin et al., 2004), Hercules Dome (Fudge et al., 2023), Dome C (Chung et al., 2023; Parrenin et al., 2017; Schwander et al., 2001) and Skytrain Ice Rise (Mulvaney et al., 2021). This model is suitable for ice-divide regions where the bed topography is mostly flat with minimal horizontal ice flow. Our model assumes pseudo-steady state conditions, meaning that the ice rise

geometry and vertical velocity shape function are stable over time (Wilchinsky & Chugunov, 2001). Basal melting is assumed to be zero because of the combination of the high SMB and thin ice.

With these assumptions, the vertical velocity $w$ relative to the bed can be expressed as:

$$w(\hat{z}) = -a\,\eta(\hat{z}) \qquad (1)$$

Here $a$ is the SMB, and $\eta$ is the shape function dependent on the non-dimensional vertical coordinate ($\hat{z} = z/H$),

with $H$ being the ice thickness and $z$ the height above the bed.

In our modelling, we consider two distinct shape functions, $\eta_1$ and $\eta_2$. In flank regions, i.e. regions far away from the ice divide, where 'divide flow' isn't dominant, the shape function $\eta_1$ can be determined as per Lliboutry (1979),

$$\eta_1(\hat{z}) = \left(1 - \frac{p+2}{p+1}(1-\hat{z}) + \frac{1}{p+1}(1-\hat{z})^{p+2}\right) \qquad (2)$$

where $p$ is a parameter for the vertical profile of deformation. For these conditions, typical $p$ values lie between 2–4 (Lliboutry,

145 1979).

Near the divide, however, the shape function $\eta_2$ is much more complicated. This is because long-term stable ice rises can have a more developed ice fabric underneath their ice divides (Martin et al., 2009a). To account for this fabric development in the divide region, we use $\eta_2$ from more sophisticated divide-flow simulations using a full Stokes thermomechanical model with a non-linear anisotropic constitutive relation between stress and strain rates (Martin and Gudmundsson, 2012).

To describe the transition between the divide flow and flank flow, we introduce a scaling parameter, $df$, that varies from 0 to 1 spatially, with one indicating that the ice flow regime is characteristic of a divide region and 0 for a typical flank flow. Thus, using $df$, the shape function $\eta$ can be determined as:

$$\eta = (1 - df)\,\eta_1 + df\,\eta_2 \qquad (3)$$

In this way, we maintain computational efficiency while complex ice deformation near the divide is prescribed.

Finally, age $\chi$ of ice in steady state, under the assumption of constant SMB through time, can be estimated as:

$$\chi(\hat{z}) = \int_z^1 \frac{1}{w(\hat{z})}\,d\hat{z} \qquad (4)$$





To ascertain the value of $df$, we implemented an optimization routine at each point on the survey lines profiles over the ice rises constrained by our knowledge of the internal stratigraphy. The procedure is as follows:

1)  Initial dating: At a point outside the divide region of the ice rise (more than four ice thicknesses), we use eq. (1), (2),
and (4) to estimate vertical profile of age. Inputs at this stage are $H$, SMB and $p$. This age-depth relation is used to date the tracked radar reflectors ($i$) at this point on the ice rise, with $\chi_i^{ini}$ the age of each reflector.

2)  Inverting for $df$: Assuming the tracked reflectors are isochronous, we determine the value of $df$ at each sample point along the reflectors using an inverse method with $\chi_i^{ini}$ as constraint. First, we use the model to determine the age of the previously dated reflectors at the sampling points and then minimize the mismatch between new age ($\chi_i^{new}$) and
initial reflector age ($\chi_i^{ini}$) using a cost function $S$:

$$S = \sum\left(\chi_i^{new} - \chi_i^{ini}\right)^2 \qquad (5)$$

3)  Monte-Carlo Scheme: We iterate steps 1–2 for $n$ times and determine $df$. For each iteration, the initial age sampling (step 1) is done at different sites outside the divide region, with different values of input $p$ randomly determined between 2–4. This way, we propagate the uncertainty in the initial age of the reflector into our $df$ estimates.

4) Final age estimate: Lastly, using the values of $df$ and eq. (3), we compute the value for η and use it to estimate age at any of the points along the survey profile. For these final age estimates, we use a constant value of 3 for $p$.

## 3. Results: Site characteristics

### 3.1. Surface and bed topography

The summit of KAM is situated at an elevation of 386 m (relative to WGS84 ellipsoid) with 525 m-thick ice below (Fig. 2a). VER's summit is slightly lower at 337 m, but it rests upon a thicker layer of ice of 565 m (Fig. 2b). Both ice rises are entirely grounded below the sea level. KAM's summit lies on a gentle southward bed slope (approximately 6 m elevation gradient per km), ~140 m beneath the WGS84 ellipsoid (or ~155 m below sea level) (Fig. 2c). This relatively flat region spans about 4 km around the summit, with a steeper slope 7 km southeast of the summit, continuing along the ridge (north-south) and towards
the survey's edge. In contrast, VER's summit is grounded deeper at around 230 m below the WGS84 ellipsoid (or ~245 m below sea level) (Fig. 2d). The region south-southwest of the summit is rather flat (~4 m elevation gradient per km), which transitions to a much steeper slope (~18 m elevation gradient per km) just past the profile going across the summit. This slope continues farther down till the extent of our measurements.

The main difference between the bed topography under the two ice rises is that KAM has a relatively flat bed in the
vicinity of the summit, whereas VER has a steep slope just at the summit. Bilinear interpolation was used to grid the bed elevation measurements (Figs. 2c and 2d). While local map topography may depend on the interpolation method, the key elevation patterns are robust, as they are directly supported by the radar data.





## 3.2 Surface Mass Balance

The dated shallow radar reflectors are used to map the spatial variability of SMB over the ice rises (Fig. 3a). Over KAM, the
reflector, dated 30 years prior, demonstrates a spatially averaged SMB of 0.29 $m_{ieq}$ $a^{-1}$. As typically observed over elevated
features like ice rises, we found the signature of a strong orographic effect visible as higher SMB on the upwind eastern side
(Lenaerts et al., 2014) and lower SMB on the downwind western side (Fig 3c). As an example, along the profile across the
KAM summit, the SMB values at 5 km on either side of the summit are 0.4 $m_{ieq}$ $a^{-1}$. (east, upwind) and 0.25 $m_{ieq}$ $a^{-1}$ (west,
downwind). Along the ice divide, roughly perpendicular to most radar profiles, a band of low SMB is seen with a
complimentary band of high SMB downwind from the ridge. This feature extends up to 8 km along the ridge from the summit,
where the ridge gets less distinct and reaches a saddle. However, the absence or weakening of these bands towards and beyond
the saddle could also be due to a lack of across-ridge measurements there.

VER has a higher spatially-averaged SMB of 0.46 $m_{ieq}$ $a^{-1}$, representative of the last 22 years. The SMB distribution
is similar to KAM in the sense of upwind-downwind contrast (Fig. 3d), with SMB values of 0.52 $m_{ieq}$ $a^{-1}$ (east, upwind) and
0.34 $m_{ieq}$ $a^{-1}$ (west, downwind) at 5 km distance on either side of the summit. There is a similar local band of minimum SMB
along the ridge, with a complimentary band of high SMB on the downwind side.

To assess the spatial representativeness of a potential millennia-scale SMB record from a deep ice core from these
ice rises, we compared the decade-scale summit SMB from the firn cores with the spatial SMB estimates from the radar data.
For this comparison, we first addressed the sampling bias inherent to the design of the radar survey by resampling the radar
data onto a 50×50 m regular grid, similar to Cavitte et al. (2022, 2023). We found that as the radial distance increases, the
mean difference between summit SMB and area-averaged SMB within this zone increases (Fig. 4a). This difference remains
minimal within 1–2 ice thicknesses from the core site, reflecting the high variability near the summit/ice divide (Fig 3a, d). At
5 km from the summit, VER and KAM show mean SMB deviations of ~21% and ~14%, respectively. This is lower than the
earlier surveyed DJU ice rise, where a deviation of 55% was found (Cavitte et al., 2023), indicating that ice-core-derived SMB
at VER and KAM are more representative of area-averaged SMB at this distance. However, this result may have been impacted
by non-even survey profile distributions at DJU (Fig. 4b, c, d). This type of fractional deviations can be considered as correction
factors when comparing local ice core SMB to regional climate models which have typical grid resolutions of 5–30 km.

## 3.3 Englacial Stratigraphy

The deep-sounding radargrams show the englacial stratigraphy along with the ice-bed interface. On the profiles across the
summits/ridges of these ice rises, we identify long-term stable englacial features known as Raymond arches (Fig. 2a, b)
(Raymond, 1983). These arches are visible in all cross-ridge profiles over KAM. At VER, these arches are visible in all cross-
ridge profiles and are absent in the region southwest of VER's summit, which is not characterized as divide flow.

In a theoretical steady state without any asymmetric external forcing, these arches are aligned vertically at the ice-
divide position over a flat bed. However, they may be tilted or advected away depending upon the speed of divide migration



relative to a characteristic timescale defined by ice thickness divided by SMB (Martin et al., 2009b). Modelling suggests that it takes one characteristic timescale for Raymond Arches to develop, and they may further evolve into a more developed double-arched shape after four characteristic timescales (e.g. Martin et al., 2009a). We used these diagnostic measures to assess the past evolution of the ice rises. The characteristic timescale for KAM is about 1800 years, and for VER, it is about 820 years; thus, VER is a more dynamic ice rise. For KAM, the Raymond Arches are vertically aligned within 1 km of the summit

and do not show a double-arch shape near the bed. This suggests no significant differential changes on either flank of KAM's summit position and, thus a stable summit position for 1–4 times its characteristic time. For VER, Raymond arches under the summit are inclined away from the summit towards the east. This observed incline can be caused by a westward migrating summit position because of differential mass balance between the flanks or the effect of the locally steeping bed near VER's summit. Lastly, the possibility of uniform thinning or thickening over the ice rises cannot be ruled out based on this analysis,

as it does not involve a change in the divide position.

### 3.4 Suitability as an ice coring site

Looking at similarities, both KAM and VER are aligned along the same longitude and are of comparable geometry. Both ice rises are similarly elevated and thick around their summits with differences <60 m (Table 1). The high surface elevation of both ice rises causes relatively cold climates with minimal surface melting, as confirmed by the lack of melt features in the

firn cores from their summits. However, they differ in their proximity to the sea, which could be the reason for the significant difference in their SMB, with the spatially mean SMB on VER greater than KAM by ~50%. The most critical differences in the context of coring site selection bed topography and englacial stratigraphy. In contrast to the steeply sloping bed close to the VER's summit, as well as the tilt in its Raymond arches, KAM's summit is placed on a relatively smooth bed and has more vertically aligned Raymond arches. Given KAM's high suitability under site criteria (I) and (II), we tentatively conclude that

KAM has a higher potential than VER as a deep ice-core site in this region. We further explore this through detailed ice-flow modelling in the following sections.

### 4 Results: Age modelling

Having evaluated the ice rises' glaciological settings, we now estimate the age of the ice at depth. First, we estimate the age of

the ice and the expected resolution of an ice core at the summits of KAM, VER, as well as DJU for further comparison. We then focus on the most suitable site, KAM and estimate the spatial distribution of age with depth, first along a radar profile across its summit and then extending the analysis to the entire survey grid surrounding the summit. These spatial age estimates will help pinpoint an optimum drilling location on KAM.



### 4.1 Age estimates at candidate ice rise summits

Here, we estimate the age and the resolution of a hypothetical deep ice core if it was drilled at the summits of these three ice rises. We make this comparison at a depth of $0.2H$ from their beds where the expected resolution is still practicable while the ice is sufficiently old (Fig. 5a). For the 1-D model, we assume divide flow ($df = 1$) (Section 2.2) and use the present-day SMB values for each ice rises summit (Table 1). We find that, while all three sites show potential for >10 ka old ice at this depth, KAM offers the potential for a significantly longer climate record at this depth. This is primarily due to thinner ice at DJU and

higher SMB at VER compared to KAM.

Additionally, we compared the expected temporal resolution of retrieved ice from the Last Glacial Maximum (LGM; 20 ka), regardless of the depth at which this age is reached (Fig. 5b). We find that DJU exhibits the least favourable resolution (~5 a cm$^{-1}$) at $0.12H$ above the bed, VER an intermediate resolution (~3.5 a cm$^{-1}$) at $0.12H$ above the bed, and KAM the most favourable resolution (~2.5 a cm$^{-1}$) at $0.16H$ above the bed. As for the age-at-depth estimates, these differences are primarily

caused by higher SMB at VER and thinner ice at DJU compared to KAM.

Adding to the earlier conclusion that KAM best meets the core site criteria (I) simple flow and (II) little surface melting, the modelling shows that KAM also best meets criteria (III) age and (IV) temporal resolution of a potential ice core. As all three ice rises meet criteria (V), accessibility to the core site, we conclude that KAM is the most suitable candidate to obtain long-term climate records beyond the LGM. However, VER and DJU can also be excellent ice core sites over the

Holocene period due to their high SMB and associated temporal resolutions.

### 4.2 Spatial variability of age-depth relationship over KAM

To precisely determine an ideal site for ice coring site in the summit area of KAM, we employ the mapview method to generate spatial maps of age with depth. The primary difference in this method from the site-specific modelling (Section 4.1) is the use of the mapped englacial stratigraphy to invert for the divide flow characteristics along the survey profiles and determine the

spatial value of $df$. We first implement this method on a profile going across the summit of KAM and later expand the modelling scope to the complete survey grid on KAM.

#### 4.2.1 Along the profile across the summit

For the first step, we used the uppermost five tracked reflectors (Fig. 6d) as they could be confidently tracked along all survey profiles. These reflectors were dated at $n$ random initial sampling points ($n = 120$) outside the divide region. The 30-year

averaged SMB, derived from the dated radar reflector along this profile, was utilized as the input SMB (see Fig 6a). Each of these points was assigned a distinct value of $p$, chosen randomly from a range of 2–4. These sampling locations are depicted in Figure 6b. For each dating instance, the optimization routine determined the values of $df$ along the profile while minimizing for age mismatch. The resulting $n$ estimates of $df$ along the profile are shown in Figure 6c. Using these $df$ values, the age-



depth relationship along the profile was estimated, resulting in $n$ different estimates. For this step, we used a fixed $p$ value of

three.

We then used these age-depth estimates to date all eight tracked reflectors. The mean age of these reflectors with their standard deviations, from top to bottom, are $0.47 \pm 0.04$ ka, $0.74 \pm 0.06$ ka, $1.38 \pm 0.15$ ka, $2.20 \pm 0.22$ ka, $3.44 \pm 0.37$ ka, $4.51 \pm 0.59$ ka and $7.23 \pm 1.48$ ka. We further compared the isochrones resulting from the mean of these estimates to the observed reflectors and found that our simple model can reproduce spatial variability in isochrone depth (Fig 6d). The model captures

the arch amplitude and the reflector slope on the eastern flank but fails to provide as good a match on the western flank or to capture the slight westward shift in the arches with depth.

All the 120 age-depth model results at the summit of KAM are shown in Figure 7a, along with associated likelihoods using a normal cumulative distribution function. Further, we calculate the depth profile of the temporal resolution of the ice core (Fig. 7b). The mean age and resolution profiles suggest an age of 20 ka at ~80 m above the bed at a resolution of ~2.5 a

cm$^{-1}$.

### 4.2.2 Over the survey grid

Here, we evaluate the spatial distribution of age in the vicinity of the summit by implementing our method over the survey grid within $10Hd$ radius around the summit, where $Hd$ stands for the ice thickness at the divide, 525 m (Table 1). For this setup, we define the divide region as the area within $4Hd$ from the summit and the vicinity of the southward ridge, approximated

as the area where the surface elevation is higher than 350 m (Fig. 8b). As for the profile modelling (Section 4.2.1), we first date the reflectors at 120 random locations outside the divide area (marked with blue circles in Fig. 8b) using randomly selecting $p$ values between 2–4. These dated reflectors were then used to invert for $df$ over the whole region within a $10Hd$ radius of the summit. The inverted $df$ values distinctly highlight the presence of divide flow with higher values along the ice rise's ridge (Fig. 7c). Using the $df$ values and $p = 3$, we estimate the age-depth relationship over the area within $10Hd$ around

the summit. Estimates of the age are shown at depth-slices of $0.6Hd$, $0.4Hd$ and $0.2Hd$ above the bed in Figure 8d–f. The older ice is located along the ridge of the ice rise, with the maxima very close to the summit.

## 5. Limitations and uncertainties of the age estimates

### 5.1 Uncertainties of present-day SMB

The uncertainty in radar-derived SMB can be primarily split into three key sources: (i) the dating of the firn core, (ii) the

picking of radar reflectors, and (iii) spatial variability in surface density. The uncertainty in the chronology of the two firn cores was calculated as the difference between the minimum and maximum age estimates at a particular depth point and was found to be ±1 year. For KAM, with an older dated reflector used for SMB estimation, the effect of this error is less significant. The uncertainty in the radar reflector picking method was found to be ±10 cm in ice equivalent depth. While estimating SMB, we used the same density-depth profile as measured from the summit firn cores over the whole survey. Detailed surface density



measurements over ice rises along the coast of DML (46 measurements over three ice rises) show density variations of ±2.5–7% around the mean values varying from 453–488 kg m$^{-3}$ (Goel et al., 2022). With the lack of similar ground-based observations, we assume the larger ±7% spatial variability. By aggregating these uncertainties from different sources through a root-sum-square approach, we estimated an overall uncertainty of ±8% for the radar-derived SMB estimates.

## 5.2 Modelling uncertainties

For our analysis, we use a simplified 1D ice flow model and assume a steady-state mass balance scenario. We use the radar-derived SMB representative of the recent decades and assume that it represents the longer time scale of the study. The longest accumulation record closest to our sites is from the Derwael Ice Rise, further east in DML, spanning 266 years. SMB averages over 30-year intervals vary by about ±17 % (standard deviation) from the long-term mean SMB of the ice rise. Longer records from inland DML plateau (cores B31, B32, B33 from Oerter et al., 2000), spanning about 740 years, show ±6% variability for

comparable 30-year averages. An even longer 2-ka spanning SMB record exists from the Law Dome ice rise in Wilkes Land, East Antarctica, which shows a variability of ±4% for the 30-year averages. Thus, to test the long-term sensitivity of our results to the single 30-year averaged radar-derived SMB for KAM, we re-run the model by scaling the input SMB by ±10 and ±25%. The prior cases are more plausible, while the latter is likely a more extreme case over the target 20 ka period. We find that at the summit location, for lower long-term mean SMB values (-10% and -25%), the 20-ka age ice is found at 71 m and 60 m

above the bed. For higher long-term mean SMB values (+10% and +25%), the 20-ka age ice is found at 85 m and 100 m above the bed.

      Since the survey sites have only been visited once, we do not have any field measurements of flow speed over the ice rises. Satellite-derived estimates of surface flow from image offset-tracking (Gardner et al., 2019) have too large uncertainties to capture the ice-flow pattern of these ice rises. However, field observations from comparable ice rises (Goel et al., 2022)

show that the horizontal advection is small near ice divides and increases gradually towards the flanks. A simple balance velocity estimate suggests speeds of only 2.5 m a$^{-1}$ at a 10$Hd$ distance from the summit. Further, we limit our analysis to within a 10$Hd$ radius from the summit and ignore any along-ridge flow, which is likely insignificant as the slope along KAM's ridge is minimal (2.8 m gradient per km). Moreover, a direct effect of along-ridge flow would be less developed Raymond Arches (Martin et al., 2009b) than what is observed and reproduced by our model.

Bedrock topography can strongly affect the vertical ice velocity near a divide (Kingslake et al., 2014), which our 1D model does not account for. VER's bed has some strong undulations in the summit area and was thus ruled out for further consideration (Fig 2d). For KAM, although the summit area overlies a gentle bed slope, the topography is smooth in wavelengths or amplitudes comparable with the ice thickness (Fig. 2c).

      The model can reproduce the reflectors well in the divide region in the modelled profile across KAM's summit (Fig.

6d). However, on the western flank, the model cannot reproduce the observed stratigraphy well. The observed reflectors here show a syncline feature at the side of the divide. We do not observe any significant asymmetry in the bed topography across the divide that could explain the feature, with similarly smooth bed and similar surface slopes on either side. The observed



mismatch on the western flank thus could be a result of a more complex 3D flow. A second possibility could be a gradual and systematic change in the spatial distribution of SMB with time, which is possible and has been speculated on other ice rises in the coastal DML region (Goel et al., 2018, 2022). However, both these possibilities act locally and thus should not affect our age-depth estimates near the summit.

A third possibility could be a rapid divide migration towards the east, resulting in the displacement of the Raymond Arch towards the west. However, the observed reflector patterns do not match those predicted by established divide migration models (Martin et al., 2009b), making this scenario less likely to explain our observations.

We implemented the ice flow in the summit area as a linear combination between flank and divide flow and used the fractional parameter $df$ to decide the strength of the divide flow. The spatial distribution of $df$ around the summit area (Figures 6c and 8c) shows that on the eastern side of the divide region, $df$ has values close to 0.5, suggesting that the flank flow itself cannot explain the reflectors in this region. Unlike the western flank, where our method fails to reproduce observations, the model matches observations well on the eastern flanks. The higher $df$ values on the eastern flank suggest divide-flow-like characteristics in this region.

Our method aims to minimize the uncertainties related to the model. To initially date the observed reflectors, we select points outside the divide region and thus avoid the complex flow pattern near the divide, but within the region where the negligible horizontal advection assumption holds. We choose a range of likely values for parameter $p$, expected in this region. To assess the sensitivity of our results to the location of the initial dating and the choice of the $p$, we carried out Monte Carlo simulations with varying sampling locations and $p$ values. This results in a range of $df$ values with corresponding different age estimates. The age-depth estimates and the resulting uncertainty probability distribution (Fig. 7a) show that the uncertainty is negligible in the top half of the ice above ~250 m from the bed. Below ~250 m, the uncertainty increases, likely due to the Raymond effect.

## 6. Conclusions

This study assessed the suitability of ice rises near Maitri Station in central Dronning Maud Land (DML) for deep ice core drilling. Our field survey and detailed analysis of the ice rises Kamelryggen (KAM) and Verbljud (VER) indicate that while both sites exhibit minimal surface melting, KAM's smoother and flatter bed and simpler englacial stratigraphy make it a superior candidate for recovering a long, continuous ice core extending back to the Last Glacial Maximum. In contrast, the higher surface mass balance of VER provides excellent resolution at shallower depths for reconstructing recent Holocene climate history.

To refine the selection of an optimal drilling site within KAM, we applied a simplified map-view 1D flow model constrained by field data. We find that KAM's summit has 20,000-year-old ice preserved at about 80 meters above the bed with a resolution of 2.5 yr cm$^{-1}$. Beyond this specific study, our modelling framework offers an efficient tool for assessing ice flow dynamics at other potential drilling sites.

An ice core record from KAM would be well-suited to investigate the nuanced interactions between sea ice, winds, and surface mass balance, thereby providing a comprehensive understanding of the regional climate dynamics in coastal DML. Ultimately, these insights will aid in refining global climate models and improving projections of future climate change in the Southern Ocean and beyond.

**Data availability**

All geophysical data and their derivatives are being prepared for submission to the NCPOR polar data centre (https://data.ncpor.res.in). For final archival, open data formats will be used. A DOI will be generated and included in the final version of the paper. Data access can be granted to reviewers promptly upon request.

**Author Contributions**

MT, KM, VG, and CM conceptualized the study and defined its objectives. VG, BP, and GM collected the field data. VG led
the data analysis, modelling, and interpretation with support from CM and KM. RD and LCM performed the ice-core analysis. RD and VG conducted the SMB representativity analysis presented in Section 3.2. VG prepared the manuscript with critical feedback from all co-authors.

**Competing interests**

Co-author Carlos Martin is a member of the editorial board of The Cryosphere.

**Acknowledgements**

This work was part of the MADICE (Mass balance, dynamics, and climate of the central Dronning Maud Land coast, East Antarctica) project, funded by the Ministry of Earth Sciences (MoES), Government of India and the Research Council of Norway. We acknowledge the logistical support provided by the Maitri station and Troll station during the field expedition. We acknowledge Jens Ivar Hauge, our field team guide.



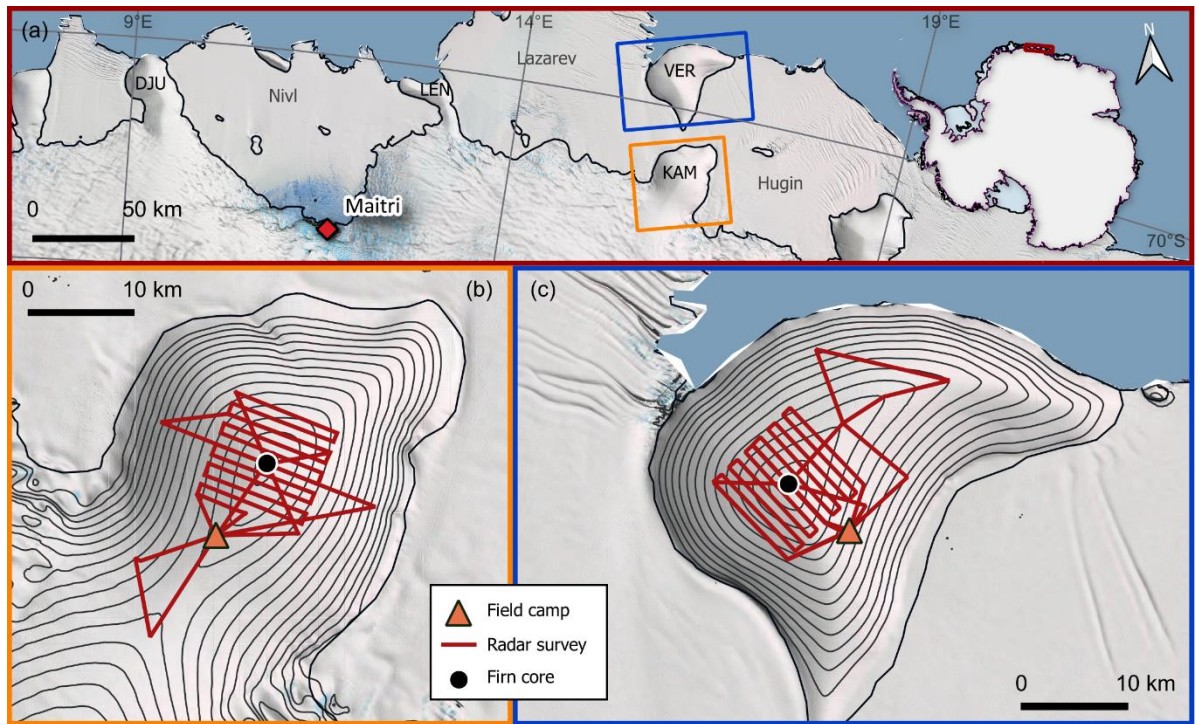

**Figure 01. Kamelryggen (KAM) and Kupol Verbljud (VER) Ice Rises, east of the Lazarev Ice Shelf, central Dronning Maud Land (DML), East Antarctica.** (a) Location of KAM and VER with black curves showing the ice sheet and ice rise grounding lines. DJU stands for Djupranen Ice Rise, and LEN stands for Leningradkollen Ice Rise. The grey text shows the names of the ice shelves. The inset shows the map coverage. The orange box over KAM shows the extent of (b), and the blue box over VER shows the extent of (c). (b–c) the zoomed-in view of ice rises with radar and firn core locations marked. The background is Landsat Image Mosaic of Antarctica (Bindschadler et al., 2008). Surface topography is shown with 20-m interval contours using the Reference Elevation Map of Antarctica (Howat et al., 2022). These figures are made in QGIS (www.QGIS.org) with polar stereographic projection using the data package Quantarctica (Matsuoka et al., 2021).





**Figure 2. Deep-sounding radargrams and bed elevation maps.** (a–b) 10-km-long deep-sounding radargram profiles going across the summits of KAM (Fig 1b) and VER (Fig 1c). The ends of these profiles (A and B) are marked in (c-d). (c–d) show the bed elevation maps for KAM and VER, respectively, with grey curves showing the location of the radar profiles used to generate these maps. The black curves show the bed elevation contours at 20 m intervals, with the labels showing the elevation below the WGS84 Ellipsoid in meters. A and B markers show the radargrams' starting and end points in (a) and (b).



**Figure 3. Shallow-sounding radargrams and SMB maps.** (a–b) show shallow-sounding radargram profiles going across the summits of KAM and VER, going from A to B. The purple curves show the tracked reflectors over the radar survey, and the white circle shows the firn core's location used to date these radar reflectors. (c–d) show the map of SMB variability (%) referenced to the spatial mean of KAM and VER, respectively, using the reflector marked in (a) and (b). A and B markers show the starting and end points of the radargrams in (a) and (b). Surface elevation contours (20 m intervals (Howat et al., 2022) and the radar profile locations (thin red curves) are shown as references.





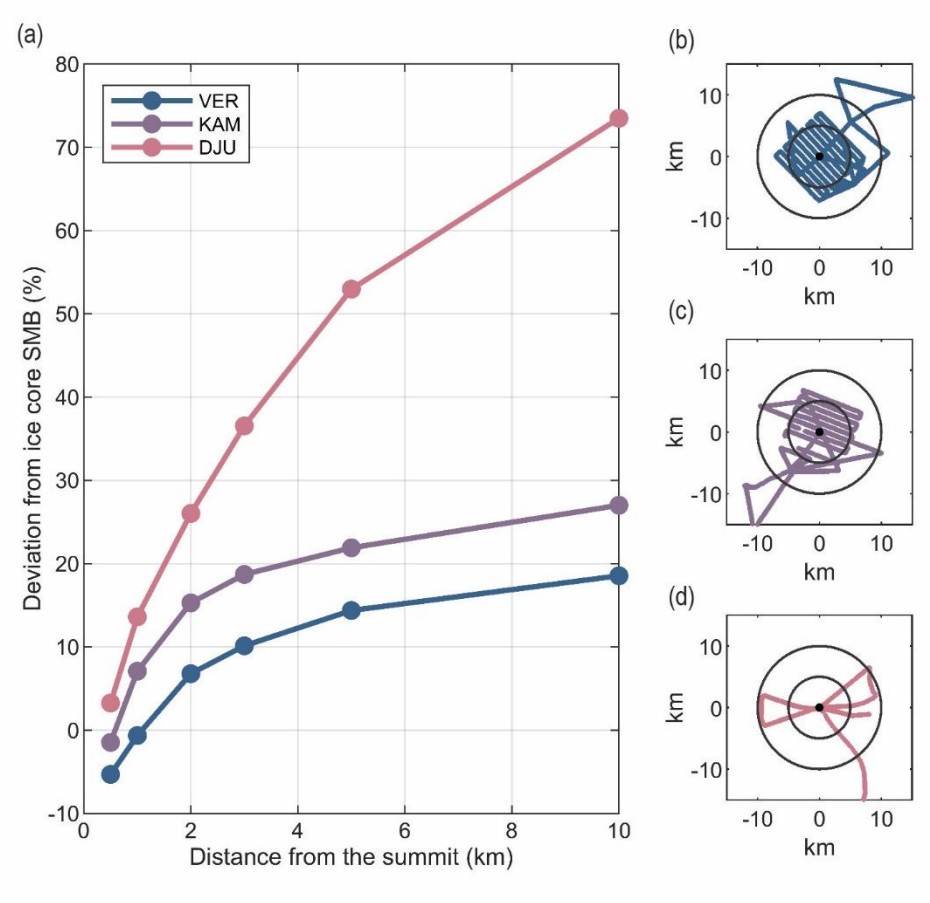


**Figure 4. Spatial representativity of the ice-core derived SMB.** Change in mean difference between ice-core derived SMB and mean radar-derived SMB with increasing radial data coverage from the VER, KAM and DJU ice-core sites. (b–d) show the coverage of the radar surveys for the three ice rises, with two concentric circles of 5 km and a 10 km radius from the summit for reference.




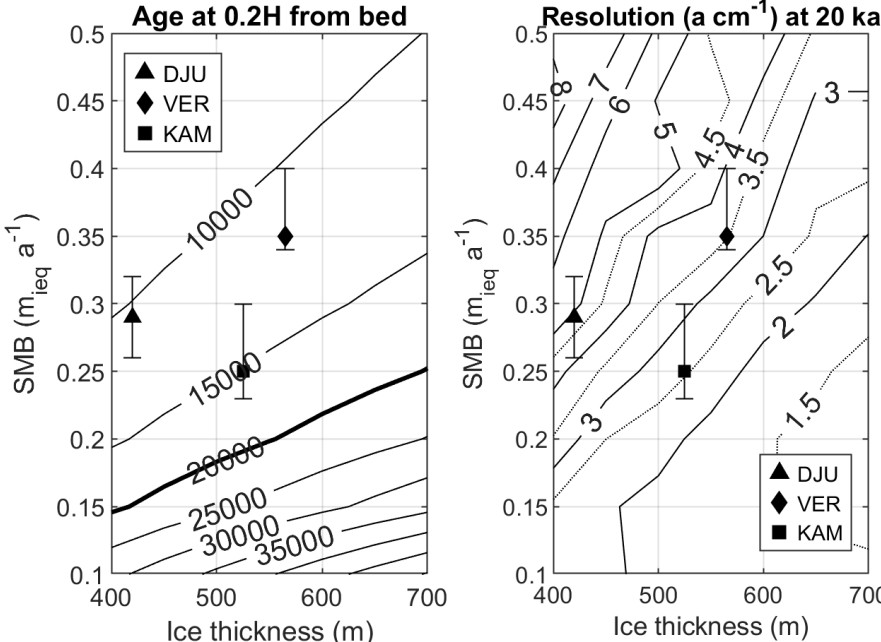

**Figure 5:** Diagrams of expected age (a) and temporal resolution (b) at 20% ice thickness above the bed in terms of ice thickness and SMB. These values at the summits of DJU, VER and KAM ice rises are also indicated. The error bars show the range of SMB within a 0.5 km radius around the summits.





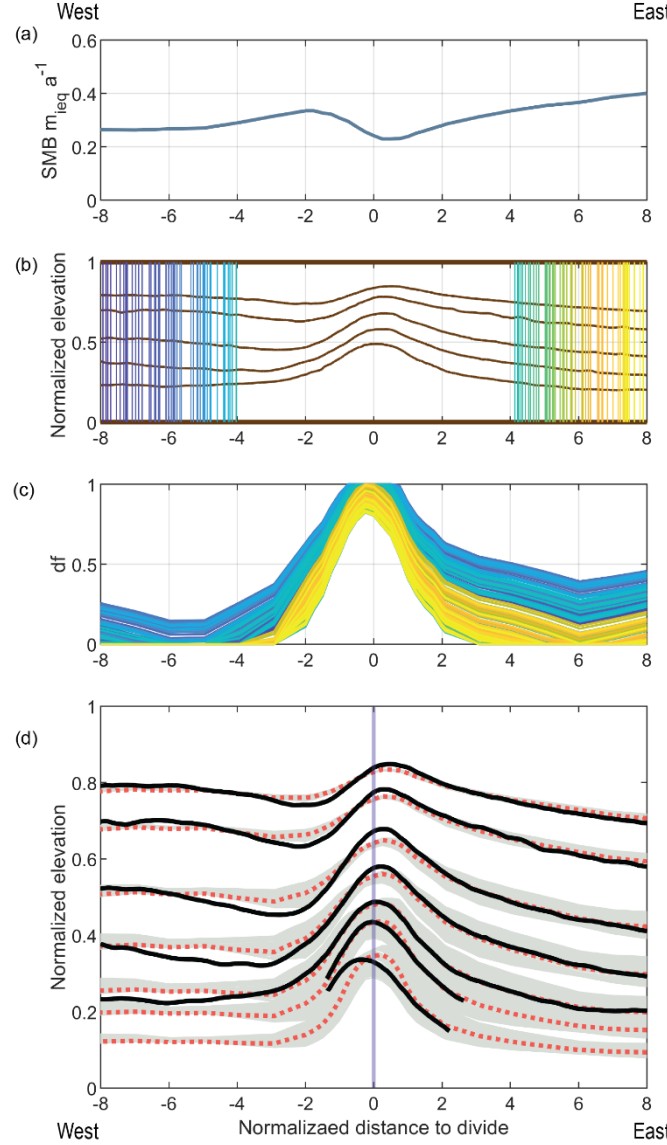


**Figure 6. Modelled ice ages along the summit profile of KAM.** (a) Input SMB estimated used a shallow radar reflector dated 30 years prior (Figure 3). (b) The tracked radar reflectors are shown as dark brown curves. The vertical lines show the 120 locations used for the initial age sampling step of the modelling routine, with a different colour for each location. These locations were randomly sampled outside of the divide region ($< 4$ km from the summits). (c) The estimated $df$ with the colours corresponding to the location of the initial sampling.

(d) The isochrones modelled using the mean $df$ (red dashed) of the 120 ensembles and the range of the isochrones modelled using the 120 ensembles $df$ values (grey), as well as the observed reflectors (black). From top to bottom, the plotted isochrones have an age of 0.47 ka, 0.74 ka, 1.38 ka, 2.20 ka, 3.44 ka, 4.51 ka and 7.23 ka. The purple vertical line shows the location of the ice divide for reference.





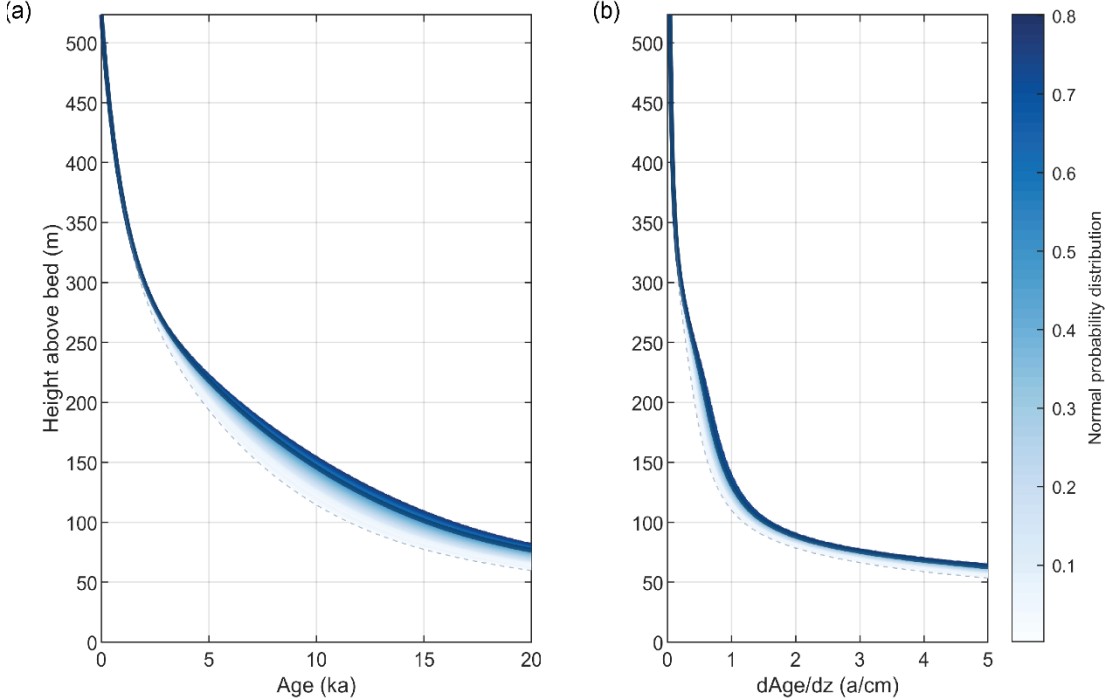

**Figure 7. Modelled age-depth relationships at the summit of KAM.** (a) Variation of age with depth at the summit of KAM. (b) Depth profile of the age resolution. The black curves in both (a) and (b) show the mean estimate, while the band shows the 120 different estimates at the location with their probability shown in colour. The dotted curves show the most extreme results.






**Figure 8: Modelling ice age in KAM's summit vicinity.** (a) Locations where the age was estimated, with magnitudes of input SMB. (b) The blue curve at the base shows the location of the available radar data. The dotted curve shows the 350 m elevation contour. The cyan colour shows the region marked as outside the divide region, with blue circles showing the location of the initial age sampling. The estimated values of parameter $df$ are shown in (c). $df$ was estimated at all locations shown in (a). Areas that do not show any markers in (c) have a very low (near zero) $df$ value. (d—f) The variation of age around the summit at different heights relative to the bed. White curves are surface elevation contours at 20 m spacing.






**Table 1.** Characteristics of VER, KAM and DJU ice rises.

| Characteristics | VER | KAM | DJU |
|---|---|---|---|
| **Traverse Distance from Maitri** | ~350 km | ~320 km | ~220 km |
| **Summit Elevation (above sea level)** | 337 m a. WGS84 ellipsoid (319 m a. s. l.) | 386 m a. WGS84 ellipsoid (369 m a. s. l.) | 337 m a. WGS84 ellipsoid (321 m a. s. l.) |
| **Ice Thickness at Summit** | 565 m | 525 m | 420 m |
| **Bed Elevation at Summit** | 228 m b. WGS84 (246 m b. s. l.) | 139 m b. WGS84 (156 m b. s. l.) | 83 m b. WGS84 (100 m b. s. l.) |
| **Ice Age at 0.2H at summit** | >12 ka | >15 ka | >10 ka |
| **Ice Resolution at 20 ka old ice at summit** | ~3.5 a cm$^{-1}$ (0.28 cm a$^{-1}$) | ~2.5 a cm$^{-1}$ (0.4 cm a$^{-1}$) | ~5 a cm$^{-1}$ (0.2 cm a$^{-1}$) |
| **Characteristic Timescale** | ~1200 a | ~1800 a | ~800 a |
| **SMB variation within 0.5 km of summit** | (0.35–0.4) m$_{ieq}$ a$^{-1}$ | (0.23–0.3) | (0.23–0.26 m$_{ieq}$ a$^{-1}$) |
| **Spatially averaged SMB** | 0.46 m$_{ieq}$ a$^{-1}$ (22 years) | 0.29 m$_{ieq}$ a$^{-1}$ (30 years) | 0.51 m$_{ieq}$ a$^{-1}$ (27 years) |
| **Bed Topography** | Steeper slope near summit | Gentle slope, flat near summit | Flat near summit, steeper slope further away |
| **Englacial Stratigraphy** | Complex, tilted Raymond Arches | Simpler, vertically aligned Raymond Arches | Complex, shifted Raymond Arches (Goel et al., 2020) |




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
