# Peer review of "A new coastal ice-core site identified in Dronning Maud Land, Antarctica, for high-resolution climate reconstructions to the Last Glacial Maximum"

_EGUsphere, 2025_

## Author Comment (AC1)

We appreciate the reviewers for investing their time and providing constructive comments on our manuscript. Overall, we revised the manuscript according to their suggestions. Below, we explain the changes we have made and present our reasoning for the suggestions we didn't follow. We hope these revisions are satisfactory and the revised manuscript meets the journal's criteria. Our responses are tabbed and in blue and follow individual comments. The line numbers we refer to are those of the marked manuscript.

**RC1: 'Comment on geosphere-2025-2037', Frédéric Parrenin, 08 Sep 2025**

Review of "A new coastal ice-core site identified in Dronning Maud Land, Antarctica, for high-resolution climate reconstructions to the Last Glacial Maximum" by Goel et al. This manuscript presents a study of several ice rises in the Dronning Maud Land region, all being accessible from the Indian Maitri station. While the DJU and LEN ice rises are briefly mentioned, the focus is then made on the more promising KAM and VER ice rises. For these two rises, a detailed radar survey has been performed. These radar surveys are used to map the SMB from a shallow horizon and firn cores at the summits. The deeper horizons show Raymond bumps, characteristic of stable ice rises.

A simple 1D age model is then fitted onto observed isochrones dated by a Lliboutry-type 1D model at the flanks where the flow is better known. From this 1D model, a 3D mapping of the age can be done and shows that KAM is the most promising site and should hold LGM ice at an acceptable resolution. I enjoyed reading this manuscript and I think it is an important contribution for glaciology and ice core science.

In my opinion, this manuscript should be accepted after a few minor corrections and improvements.

It is great to hear that the reviewer enjoyed reading our manuscript and found it worthy of publication. As described below, we revised the manuscript as guided by the reviewer's suggestions.

**General comments:**

• The age model used is said to be pseudo-steady, but I have the impression that it is just steady.

The difference between the two just comes from a change of the time variable based on SMB variations (see Parrenin et al., JG, 2006 and Parrenin and Hindmarsh, JG, 2007 for details). Here, the temporal variations of SMB could be taken into account by using, e.g., the EDML SMB temporal variations. This would affect the age and resolution of the deepest layer, close to the LGM, where SMB was probably ~2 times smaller. Numerically, this is really easy to do so I suggest to do it if it has not been done.

Thanks! We agree with your assessment. As implemented, our model is in steady-state. We considered using time-varying SMB input but since there are no reasonable continuous SMB records that could be used as an input to such a pseudo-steady state model, we instead tested multiple steady-state scenarios with scaled SMB values (±10% and ±25%). The amount of scaling was based on the few available coastal SMB records. Please refer to section 5.2 para (L327) for these discussions:

"For our analysis, we use a simplified 1D ice flow model and assume a steady-state mass balance scenario. We use the radar-derived SMB representative of the recent decades and assume that it represents the longer time scale of the study. The longest accumulation record closest to our sites is from the Derwael Ice Rise, further east in DML, spanning 266 years. SMB averages over 30-year intervals vary by about  $\pm 17\%$  (standard deviation) from the long-term mean SMB of the ice rise. Longer records from inland DML plateau (cores B31, B32, B33 from Oerter et al., 2000), spanning about 740 years, show  $\pm 6\%$  variability for comparable 30-year averages. An even longer 2-ka spanning SMB record exists from the Law Dome ice rise in Wilkes Land, East Antarctica, which shows a variability of  $\pm 4\%$  for the 30-year averages. Thus, to test the long-term sensitivity of our results to the single 30-year averaged radar-derived SMB for KAM, we re-run

the model by scaling the input SMB by  $\pm 10$  and  $\pm 25\%$ . The prior cases are more plausible, while the latter is likely a more extreme case over the target 20 ka period. We find that at the summit location, for lower long-term mean SMB values (-10% and -25%), the 20-ka age ice is found at 71 m and 60 m above the bed. For higher long-term mean SMB values (+10% and +25%), the 20-ka age ice is found at 85 m and 100 m above the bed."

• The value of the Lliboutry exponent p is taken between 2 and 4. I am not sure where these values come from, so proper references would help. From the original 1979 Lliboutry article, an estimate of p can be done using the Shallow Ice Approximation and an estimate of the temperature gradient at the base. If I remember correctly, the p value for Vostok is more around 8. Not sure what is the temperature gradient at the base here, so the value might be different.

Thank you for raising this point. You are correct that p values at deep ice core sites like Vostok are considerably higher (~8). However, our sites differ significantly in their glaciological conditions.

Following Lliboutry (1979), p is calculated as:

$$p = n - 1 + k \cdot G \cdot H$$

where  $k=Q/(R\cdot T_b^2)$  with Q=60 kJ/mol , R=8.31 J/(mol·K) , and  $T_b$  is the basal temperature [K]. The temperature gradient  $G=q_{geo}/\kappa_{ice}$ , where  $q_{geo}$  is the geothermal heat flux [W/m²] and  $\kappa_{ice}=2.1$  W/(m·K) is the thermal conductivity of ice. H is the ice thickness.

For our relatively thin ice rise ( $H \sim 525$  m) with a frozen bed, we used geothermal flux values of 40-60 mW/m² and estimated basal temperatures of 260-270 K. Under these conditions, the thermal term  $k \cdot G \cdot H$  is only slightly greater than 1, resulting in p values close to n (typically 3-4 for ice). In contrast, the thick ice at Vostok (H > 3700 m) with a temperate bed produces much larger thermal terms, hence  $p \sim 8$  as you highlighted.

Using the above stated values and a wide range of n between 2 and 5 we estimated p as shown in the figure below:

In our study we use a p range of 2-4. Given the expected range of n over ice rises, this range covers the most likely p values and is adequate for assessing sensitivity in our age-depth model, even if not optimally centered. We rephrased the text in the manuscript as (L150):

"For thin ice rises with frozen beds, typical p values lie between 2–4 (Lliboutry, 1979). This range is notably lower than deeper ice-core sites like Vostok ( $p \approx 8$ ) with significantly thicker ice and warm basal conditions (Parrenin et al., 2007)."

• From the radargrams, it seems Raymond bumps are surrounded by troughs, at least on one side. Parrenin and Hindmarsh (JG, 2007), showed that horizontal advection of the ice can create these troughs. Not sure it is the correct explanation here, but at least it could be worth mentioning.

Indeed. We agree these "troughs" (or synclines) observed on both ice rises as you point out, could be a result of horizontal advection effects among other possibilities (localized SMB changes, divide migration). We have now updated the discussion of this syncline in section 5.2 at (L351):

"The model can reproduce the IRHs well in the divide region in the modelled profile across KAM's summit (Fig. 6d). However, on the western flank, the model cannot reproduce the observed stratigraphy well. The observed IRHs here show a syncline feature at the side of the divide. We do not observe any significant asymmetry in the bed topography across the divide that could explain the feature, with similarly smooth bed and similar surface slopes on either side. The observed mismatch on the western flank thus could be a result of a more complex 3D flow. Parrenin and Hindmarsh (2007) demonstrated that spatial variations in velocity profiles, particularly an abrupt transition from dome to flank flow regimes, can generate synclines flanking Raymond arches. A second possibility could be a gradual and systematic change in the spatial distribution of SMB with time, which is possible and has been speculated on other ice rises in the coastal DML region (Goel et al., 2018, 2022). However, both these possibilities act locally and thus should not affect our age-depth estimates near the summit.

A third possibility could be a rapid divide migration towards the east, resulting in the displacement of the Raymond Arch towards the west (Nereson and Waddington, 2002). However, the observed IRH patterns do not match those predicted by established divide migration models (Martin et al., 2009b), making this scenario less likely to explain our observations."

• By the way, using a flow tube model like in Parrenin et al. (GMD, in press) and Chung et al. (TC, in press) could be a possible perspective for the modeling exercise to take into account horizontal advection.

Thank you for this suggestion. A flow tube model approach could indeed be a next step to include more physics while keeping the computation costs low as intended in this work.

**Minor comments:**

• I. 66: Maybe introduce the "LEN" notation here.

Added.

• I. 134: Not sure your model is really pseudo-steady, see comment above.

Updated to "steady state"

I. 142: "isn't" -> "is not"

Corrected.

• I. 144: "p values lie between 2-4" -> see comment above

Please see our response to the earlier comment.

 section 4.1: I am not sure to understand the comparison of this section. It is said in the beginning that the comparison will be done at 0.2H over the bed. Then the comparison is made at 0.12H for DJU and VER and 0.16H for KAM.

Apologies for the confusion. We are making two separate comparisons for age and resolution. For age we pick a fixed depth of 0.2H, while for resolution, we pick a fixed age of 20 ka. We have now revised the text to be more clear about this.

**Line 258:**

"Here, we estimate the age and the resolution of a hypothetical deep ice core if it was drilled at the summits of these three ice rises. For age comparison, we make the comparison at a depth of 0.2H from their beds where the expected resolution is still practicable while the ice is sufficiently old (Fig. 5a)...."

"For resolution comparison, we compare the expected temporal resolution of retrieved ice from the Last Glacial Maximum (LGM; 20 ka), regardless of the depth at which this age is reached (Fig. 5b)."

• Figure 2: I would rather use dark colours for troughs and light colours for highs.

It is indeed the case in the figure. Since the colorscale is in "Bed elevation (m below WGS84 ellipsoid)', the lighter color indicate shallower regions (highs) and darker colors indicate the troughs. The figure is thus unchanged.

• I. 407: Should not is be Fig. 1a and 1b instead of Fig. 1b and 1c?

We want to refer to the maps in the figure1 focusing the individual ice rises. Fig 1a is the regional map, while 1b focuses on KAM and 1c focuses on VER. We have left this caption unchanged.

• Figure 3: The labels of the sub-figures do not seem to correspond to the legend. They are also ordered from top to bottom, which is not consistent with Figure 2.

Thank you for pointing this out. We have now revised the figure and the associated references in the manuscript.

• Figure 6d: There is a model-obs discrepancy, which could be due to horizontal advection (see comment above) or to non-steady features such as varying accumulation pattern.

Correct. As mentioned in our response to the previous comment, we have discussed this discrepancy in section 5.2 (L351).

---

## Author Comment (AC2)

We appreciate the reviewers for investing their time and providing constructive comments on our manuscript. Overall, we revised the manuscript according to their suggestions. Below, we explain the changes we have made and present our reasoning for the suggestions we didn't follow. We hope these revisions are satisfactory and the revised manuscript meets the journal's criteria. Our responses are tabbed and in blue and follow individual comments. The line numbers we refer to are those of the marked manuscript.

**RC2: 'Comment on egusphere-2025-2037', Anonymous Referee #2, 06 Oct 2025**

This article explores the merits of potential ice core sites on 2 ice domes in the Dronning Maud Land region of Antarctica, for extracting an ice core with a high resolution climatic record going back to the Last Glacial Maximum (~20 ka). They present shallow and deep radar surveys over both areas. They carried out a detailed assessment of the topographical setting, deeming the Kamelryggen (KAM) ice dome to be more suitable than Kupol Verbljud. They then applied a simple age-depth model would accounted for both dome and flank ice flow. The model determined the likely age and resolution of ice through the thickness of the ice sheet, concluding that 20 ka ice may be around 80 m above the bed at KAM making it a suitable ice core drill site. The study is detailed and thorough, assessing the area through both observations and modelling. The results and arguments are clearly presented, I especially appreciated Table 1 for easy direct comparison of the ice domes. Therefore, I recommend this paper for publication with a few changes.

We are pleased to hear that the reviewer found our manuscript thorough and found it worthy of publication. We are thankful for all the constructive comments. As described below, we revised the manuscript as guided by the reviewer.

**Specific comments**

• My first comment relates to the use of the word "reflector". Here, the term "internal reflection horizon (IRH)" is commonly used as in the AntArchitecture review covering radar stratigraphy in Antarctica (Bingham et al., accepted). Therefore using "IRH" could help the community move to more standardised language when referring to these radar phenomena.

Thank you for bringing this up. Sometimes it takes a little nudging to move away from what we are used-to to what the community has agreed upon. We have now revised the manuscript to replace the word "reflector" with "IRH".

• It would be helpful to explain fairly early on, that there is a single IRH tracked in the shallow radar used for determining SMB. Then there are several other IRHs presumably tracked in the deep radar used for age-depth profiling. I found it quite difficult to disentangle at various points in the manuscript as they are often just referred to as "the reflector(s)".

Thank you for bringing up this issue. We have now revised the manuscript to clarify which IRHs were tracked from the two datasets.

L125:

"To study the spatial distribution of SMB over the ice rises, we tracked the deepest prominent radar reflector visible in the shallow radar stratigraphy that could be dated using the firn core. .. Additionally, we tracked seven reflectors visible in the deep radar stratigraphy of KAM (approx. depths of  $\sim$ 0.84H,  $\sim$ 0.73H,  $\sim$ 0.65H,  $\sim$ 0.58H,  $\sim$ 0.49H,  $\sim$ 0.43H and  $\sim$ 0.33H at the summit) with the top five continuously tracked over the complete survey, while the bottom two limited to the vicinity of the ice divide."

• The IRHs in Fig 6d are never fully introduced. From their ages I assume they must be from the deep radar system. There should be a paragraph describing how many IRHs were tracked in total, at what depth/fraction of ice

thickness, with which radar system and referencing a figure so the reader can see the coverage. Such descriptions could be added at the end of section 2.1.

**Please refer to the previous comment.**

- It should then be made clearer throughout the manuscript, whether you are referring to the shallow or deep IRHs. Eg.
  - L159-171 mention that it is the deep IRHs used for this purpose

**Corrected.**

 L273 - make it clear that here you are talking about the deep IRHs. As in the next sentence you talk about the shallow IRH at KAM

**Corrected. Rephrased at L276 as:**

"The primary difference in this method from the site-specific modelling (Section 4.1) is the use of the mapped englacial stratigraphy (i.e. tracked deep IRHs) to invert for the divide flow characteristics along the survey profiles and determine the spatial value of df."

• L278 - age mismatch with the deep IRHs.

**Corrected.**

• L281 - Mentions "eight reflectors" but there are only 7 in Fig 6a. Would also be good to reference Fig 6a here.

**Corrected.**

• Figure 7 - There are a few things I could not make out in this figure, even while zooming in on the pdf. I could not see a black curve showing the mean estimate. Perhaps it is the thick dark blue curve that does not follow the colour gradient? In which case it would be good to change this to a thinner black line.

Thank you for bringing this up. The dark curve was indeed the mean. Have updated this in the new figure and included a legend.

• My other comments are based on this assumption. Does the mean age and resolution mean the most probable? Or is it the weighted average age from your 120 results taking into account each profile's likelihood. This should be further clarified in the text (L287-290).

To clarify: this is the unweighted mean of all the 120 ensemble members with each ensemble given equal weight. Each of the ensemble member represents a different plausible scenario through which we propagate the expected uncertainty in our age estimates. The text has been rephrased for clarity (please see our response to the next point.)

• Ignoring the dark blue line which I think may be the mean, it seems as though the gradient (in Fig 7a), and therefore probability, increases towards the upper bound ie. where older ice is shallower. As the probability increases but does not seem to reach a maximum, why were scenarios not tested where the older ice is even shallower?

The figure showed the normalized cumulative density function of the ensemble age estimates with each horizontal slice. So at each point along a depth it gave the cumulative probability, showing the fraction of results that lie to the left of that point. The reviewer's comment prompted us to reconsider our

visualization approach assuming normal distribution, and instead opting for a distribution-free representation as explained below.

In this specific case of age-sampling at the summit, where we expect the ice flow to show "divide flow" characteristics i.e., with the df approaching 1, the distribution is not normal but skewed. To represent these results without assuming any predefined distribution pattern we adopt the method of showing  $\pm \sigma$  instead, where  $\sigma$  represents the standard deviation-equivalent percentile intervals: 68% interval (16th-84th percentiles, approximately  $\pm 1\sigma$ ) and 95% interval (2.5th-97.5th percentiles, approximately  $\pm 2\sigma$ ). Percentile bands are distribution-free, with percentiles derived by sorting the ensemble members and selecting values at specific positions. The squeezing of the bands towards the right suggests the clustering of the age estimates towards divide flow characteristic estimate. The revised plot also show the min-max values to show the range of age estimates as well as the mean and mode locations.

Also revised the text at Section 4.2.1 Para 3 (L296) as:

"Age-depth estimates at the summit are shown in Figure 7a with empirical percentile intervals: 68% interval (16th-84th percentiles, approximately  $\pm 1\sigma$ ) and 95% interval (2.5th-97.5th percentiles, approximately  $\pm 2\sigma$ ), alongside ensemble mean and median. At the summit mean and median diverge, indicating skewness in the ensemble results, as expected at the ice divide where df approaches one. Further, we calculate the depth profile of the temporal resolution of the ice core (Fig. 7b). The mean age and resolution profiles suggest an age of 20 ka at ~80 m above the bed at a resolution of ~2.5 a cm-1."

• It may be useful to include an inset in the top right hand corner to zoom in on the deeper section, as this would make the colour gradient and lines easier to differentiate by eye.

We acknowledge the reviewer's concern about visual clarity, and found it more informative to include a histogram of ensemble results at a specific depth slice to understand the distribution better. We also include the mean/median location from the plot in this histogram.

**Minor Corrections**

• L16 - A radar survey reveals...

**Corrected.**

L24 - ~ 20 ka

**Corrected.**

• L32 - "... often exhibit distinct characteristics from comparable events...". This sentence is unclear as is sounds as though event characteristics which appear very clearly in the Northern Hemisphere, also appear in the Southern Hemisphere climate records. But I think you mean that these events can have different characteristics in Southern Hemisphere records, so this should be reworded to make it clearer.

**Corrected. Now rephrased as L(31):**

"These events, such as the Last Deglaciation (18-11 ka; Mayewski et al., 1996), the Antarctic Cold Reversal (14.7-13 ka; Pedro et al., 2016), and the abrupt cooling event 8.2 ka ago (Stager and Mayewski 1997), often display characteristics that contrast with those observed in comparable Northern Hemisphere events."

• L48 - sea ice variability

Corrected.

L58 - the Antarctic coast

Corrected.

• L104 - add a reference for the value of 169 m us-1

Added.

• L131 - the ice's age

Corrected.

• L134 - from your description, it seems to be a steady model, not pseudo-steady but reviewer 1 has already mentioned this in more detail

Corrected.

• L183 - "...until the extent of ..."

Corrected.

• L185 - "...VER has a steep slope just at the summit." From looking at Fig 2d, it is clear what you are referring to but the sentence alone is unclear. It could be changed to something like "... the summit of VER occurs at the edge of a steep slope in the bed."

Corrected.

• L190 - "30 years prior" - this phrasing sounds like age of the reflector was determined 30 years before the current study was carried out, so maybe change to " the age of the reflector is 30 a", if that is the intended meaning or clarify further if not.

Corrected.

• L236 - "mean spatial" instead of "spatially mean"

**Corrected.**

• Also L236 - "The most critical differences ... are bed topography ..."

**Corrected.**

• L276 - if I understand correctly, this random value of p is then optimised? So it may be helpful to remind the reader by saying "... assigned a distinct initial value of p..."

To clarify: the random p values (2-4 range) are not optimized. Each ensemble run uses its assigned p value to calculate initial ages at locations outside the divide region. Then, at each spatial location, we optimize the df to reproduce these initial age estimates. The resulting df distribution thus incorporates the uncertainty associated with the choice of p in the initial sampling step.

• L299 - Figure 7c does not exist

**Corrected.**

Reference Bingham, R. G. et al. (Accepted). Antarctica's internal architecture: Towards a radiostratigraphically-informed age-depth model of the Antarctic ice sheets. EGU Preprint Repository, 17, 33. https://doi.org/10.5194/egusphere-2024-2593

---

## Author Comment (AC3)

**A new coastal ice-core site identified in Dronning Maud Land, Antarctica, for high-resolution climate reconstructions to the Last Glacial Maximum**

Vikram Goel1, Carlos Martin2, Kenichi Matsuoka3, Bhanu Pratap1, Geir Moholdt3, Rahul Dey1, Chavarukonam M. Laluraj1, Meloth Thamban1

10

Correspondence to: Vikram Goel (vikram.goel@outlook.com)

Abstract. High-resolution ice cores from the Antarctic Ice Sheet margin are crucial for reconstructing the climate history of Antarctica and the Southern Ocean. Ice-rise summits with stable positions and substantial snow accumulation can be ideal sites for such ice cores. We surveyed two ice rises at 16° E, at the eastern edge of the Lazarev Ice Shelf. Kupol Verbljud (VER) is an isle at the calving front, and Kamelryggen (KAM) is a promontory landward of VER. RadarThe radar survey reveals ice thicknesses of 560 m under VER's summit and 525 m under KAM's summit. The long-term stable englacial features, Raymond Arches, are observed in both ice rises, but while VER's arches are tilted, KAM exhibits vertically-aligned arches within its summit, indicating a more stable summit position. We find KAM's summit area better suited for a long ice core, given its gentler bed slope and simpler ice stratigraphy. Surface mass balance derived from dated reflectors internal reflection horizon show consistent spatial patterns over recent decades. Using a one-dimensional age-depth model we consider the local ice flow as a combination of two extreme cases: diverging divide flow and shear-dominated flank flow. We determine which combination of these flow regimes best reproduces the mapped englacial radar stratigraphy and use it to estimate the age of ice. We conclude that KAM's summit is well-suited for obtaining a high-resolution ice core record beyond the Last Glacial Maximum with expected ~20 kiloyearka-old ice at a depth 80 m above the bed where the resolution is expected to be 2.5 a cm-1.

<sup>1National Centre for Polar and Ocean Research, Ministry of Earth Sciences, Goa, India

<sup>2British Antarctic Survey, Natural Environmental Research Council, Cambridge, UK

<sup>3Norwegian Polar Research Institute, Norway

**30 1 Introduction**

35

40

55

60

Antarctica has experienced several significant climatic events since the Last Glacial Maximum (LGM; 20 ka). These events, such as the Last Deglaciation (18-11 ka; Mayewski et al., 1996), the Antarctic Cold Reversal (14.7-13 ka; Pedro et al., 2016), and the abrupt cooling event 8.2 ka ago (Stager and Mayewski 1997), often exhibit distinct display characteristics from that contrast with those observed in comparable events in the Northern Hemisphere events (
[revised manuscript text omitted]
 IRH visible in the shallow radar stratigraphy and that could be dated it—using the firn core- (depth at summit ~10 m for both ice rises). Because the tracked reflector IRH is shallow (the ratio of the reflector IRH depth and ice thickness is approximately 0.03), vertical strain effects on its depth are negligible (the shallow layer approximation, Waddington et al., 2007). To estimate

the mass above the reflector IRH, we assume no lateral variations in the vertical firn density profile and use the density profiles from the firn cores at each ice rise. Dividing this mass by the reflector's age provides the SMBIRH's age provides the SMB.

Additionally, we tracked seven IRHs visible in the deep radar stratigraphy of KAM (approx. depths of ~0.84H, ~0.73H, ~0.65H, ~0.58H, ~0.49H, ~0.43H and ~0.33H at the summit) with the top five continuously tracked over the complete survey, while the bottom two limited to the vicinity of the ice divide.

**2.2. Ice-flow model**

135

155

160

To estimate the ice's age of ice around the ice rise summits, we used a simplified 1-D map view age-depth model along the radar surveys. Similar simplified models have been used to estimate ice's age at other drilling sites like Vostok (Parrenin et al., 2001; Parrenin et al., 2004), Hercules Dome (Fudge et al., 2023), Dome C (Chung et al., 2023; Parrenin et al., 2017; Schwander et al., 2001) and Skytrain Ice Rise (Mulvaney et al., 2021). This model is suitable for ice-divide regions where the bed topography is mostly flat with minimal horizontal ice flow. Our model assumes pseudo-steady state conditions, meaning that the ice rise geometry and vertical velocity shape function are stable over time (Wilchinsky & Chugunov, 2001). Basal melting is assumed to be zero because of the combination of the high SMB and thin ice.

With these assumptions, the vertical velocity w relative to the bed can be expressed as:

$$w(\hat{z}) = -a \, \eta(\hat{z}) \tag{1}$$

Here a is the SMB, and  $\eta$  is the shape function dependent on the non-dimensional vertical coordinate ( $\hat{z} = z/H$ ), with H being the ice thickness and z the height above the bed.

In our modelling, we consider two distinct shape functions,  $\eta_1$  and  $\eta_2$ . In flank regions, i.e. regions far away from the ice divide, where 'divide flow' isn't is not dominant, the shape function  $\eta_1$  can be determined as per Lliboutry (1979),

$$\eta_1(\hat{z}) = \left(1 - \frac{p+2}{p+1}(1-\hat{z}) + \frac{1}{p+1}(1-\hat{z})^{p+2}\right) \tag{2}$$

where p is a parameter for the vertical profile of deformation. For these conditions thin ice rises with frozen beds, typical p values lie between 2–4 (Lliboutry, 1979). This range is notably lower than deeper ice-core sites like Vostok ( $p \approx 8$ ) with significantly thicker ice and warm basal conditions (Parrenin et al., 2007).

Near the divide, however, the shape function  $\eta_2$  is much more complicated. This is because long-term stable ice rises can have a more developed ice fabric underneath their ice divides (Martin et al., 2009a). To account for this fabric development in the divide region, we use  $\eta_2$  from more sophisticated divide-flow simulations using a full Stokes thermomechanical model with a non-linear anisotropic constitutive relation between stress and strain rates (Martin and Gudmundsson, 2012).

To describe the transition between the divide flow and flank flow, we introduce a scaling parameter, df, that varies from 0 to 1 spatially, with one indicating that the ice flow regime is characteristic of a divide region and 0 for a typical flank flow. Thus, using df, the shape function  $\eta$  can be determined as:

$$\eta = (1 - df) \, \eta_1 + df \, \eta_2 \tag{3}$$

In this way, we maintain computational efficiency while complex ice deformation near the divide is prescribed.

Finally, age  $\gamma$  of ice in steady state, under the assumption of constant SMB through time, can be estimated as:

$$\chi(\hat{z}) = \int_{z}^{1} \frac{1}{w(\hat{z})} d\hat{z} \tag{4}$$

To ascertain the value of df, we implemented an optimization routine at each point on the survey lines profiles over the ice rises constrained by our knowledge of the internal stratigraphy. The procedure is as follows:

- 1) Initial dating: At a point outside the divide region of the ice rise (more than four ice thicknesses), we use eq. (1), (2), and (4) to estimate vertical profile of age. Inputs at this stage are H, SMB and p. This age-depth relation is used to date the tracked radar reflectors deep IRHs (i) at this point on the ice rise, with  $\chi_i^{ini}$  the age of each reflector IRH.
- 2) Inverting for df: Assuming the tracked reflectors IRHs are isochronous, we determine the value of df at each sample point along the reflectors IRHs using an inverse method with  $\chi_i^{\text{ini}}$  as constraint. First, we use the model to determine the age of the previously dated reflectors IRHs at the sampling points and then minimize the mismatch between new age  $(\chi_i^{\text{new}})$  and initial reflector IRH age  $(\chi_i^{\text{ini}})$  using a cost function S:

$$S = \sum (\chi_i^{\text{new}} - \chi_i^{\text{ini}})^2$$
 (5)

[revised manuscript text omitted]

Figure 7. Modelled age-depth relationships and ice-core resolution at the summit of KAM. (a) Variation of age with depth at the summit of KAM. (b) Depth profile of the age-resolution. The black curves in both (a) and (b) show the cm-1) versus height above bed. Each showing ensemble mean estimate, while the band shows the 120 different estimates at the location with their (solid black line), median (dashed black line), and empirical probability intervals (68% interval: 16th-84th percentiles, approximately ±1σ; 95% interval: 2.5th-97.5th percentiles, approximately ±2σ, shaded from dark to light blue). Min/max values shown in colour. The dotted curves show the most extreme results. The black horizontal line at 100 m above bed in (a) indicates the depth sampled for the inset histogram, which shows the distribution of age estimates at that depth with corresponding mean (red) and median (black) marked.

Figure 8: Modelling ice age in KAM's summit vicinity. (a) Locations where the age was estimated, with magnitudes of input SMB. (b) The blue curve at the base shows the location of the available radar data. The dotted curve shows the 350 m elevation contour. The cyan colour shows the region marked as outside the divide region, with blue circles showing the location of the initial age sampling. The estimated values of parameter df are shown in (c). df was estimated at all locations shown in (a). Areas that do not show any markers in (c) have a very low (near zero) df value. (d—f) The variation of age around the summit at different heights relative to the bed. White curves are surface elevation contours at 20 m spacing.

**Table 1.** Characteristics of VER, KAM and DJU ice rises.

[revised manuscript text omitted]